# Catalytic Effect of Facile Synthesized TiH_1.971_ Nanoparticles on the Hydrogen Storage Properties of MgH_2_

**DOI:** 10.3390/nano9101370

**Published:** 2019-09-24

**Authors:** Liuting Zhang, Xiong Lu, Liang Ji, Nianhua Yan, Ze Sun, Xinqiao Zhu

**Affiliations:** 1School of Energy and Power, Jiangsu University of Science and Technology, Zhenjiang 212003, China; 152210804218@stu.just.edu.cn (X.L.); 179210016@stu.just.edu.cn (L.J.); 189210007@stu.just.edu.cn (N.Y.); 182210019@stu.just.edu.cn (Z.S.); 2Institute of Nuclear Physics and Chemistry, China Academy of Engineering Physics, Mianyang 621999, China

**Keywords:** hydrogen storage, MgH_2_, TiH_1.971_, catalytic effect, reversibility

## Abstract

Catalytic doping plays an important role in enhancing the hydrogen storage performance of MgH_2_, while finding an efficient and reversible catalyst remains to be a great challenge in enhancing the de/rehydrogenation properties of MgH_2_. Herein, a bidirectional nano-TiH_1.971_ catalyst was prepared by a wet chemical ball milling method and its effect on hydrogen storage properties of MgH_2_ was studied. The results showed that all the TiH_1.971_ nanoparticles were effective in improving the de/rehydrogenation kinetics of MgH_2_. The MgH_2_ composites doped with TiH_1.971_ could desorb 6.5 wt % H_2_ in 8 min at 300 °C, while the pure MgH_2_ only released 0.3 wt % H_2_ in 8 min and 1.5 wt % H_2_ even in 50 min. It was found that the smaller the size of the TiH_1.971_ particles, the better was the catalytic effect in promoting the performance of MgH_2_. Besides, the catalyst concentration also played an important role and the 5 wt %-*c*-TiH_1.971_ modified system was found to have the best hydrogen storage performance. Interestingly, a significant hydrogen absorption amount of 4.60 wt % H_2_ was evidenced for the 5 wt %-*c*-TiH_1.971_ doped MgH_2_ within 10 min at 125 °C, while MgH_2_ absorbed only 4.11 wt% hydrogen within the same time at 250 °C. The XRD results demonstrated that the TiH_1.971_ remained stable in cycling and could serve as an active site for hydrogen transportation, which contributed to the significant improvement of the hydrogen storage properties of MgH_2_.

## 1. Introduction

The increase of pollutants such as nitrogen dioxide in the atmosphere, resulted from high consumption of fossil fuels, causes an urgent demand for clean and sustainable energy resources [1]. Among various clean energy sources, hydrogen is considered to be one of the most promising energy carriers because of its environmental friendliness and high energy content (142 MJ/kg) [2,3]. Nevertheless, hydrogen storage technology remains a challenge for the widespread use of hydrogen energy [4,5]. So far, the solid hydrogen storage technology has become a research hotspot benefiting from its high safety and convenient transportation where hydrogen is stored by adopting physical method via intermolecular forces (metal-organic frameworks (MOFs) and carbon) or chemical method through chemical bonds (complex hydrides and metal hydrides) [6,7,8,9,10]. At present, MgH_2_ shows vast potential to be used as a hydrogen storage material because of its abundant reserves, low cost, large volumetric (>100 kg/m^3^), and gravimetric densities (> 7.6 wt %H_2_) [11,12,13]. The Mg–H bond is very stable and difficult to break because of the thermodynamic stability of MgH_2_ [14], leading to higher dehydrogenation temperature (300~400 °C) [15,16,17]. Besides, the slower kinetic performance (1 wt % H_2_/min, at 300 °C) is another tough challenge [18]. In order to deal with problems mentioned above, researchers worldwide tried oceans of modification technics to improve the hydrogen storage performance of MgH_2_, including nanostructuring, surface modification, alloying, destabilization, and doping with transition metals [19,20,21,22,23,24,25,26,27,28].

Especially, adding transition metals and related compounds such as ZrMn_2_, Ni/CMK-3, TiO_2_@C, Ta_2_O_5_, and Li_2_TiO_3_ proved to be one of the highly efficient methods in improving the dynamic performance of MgH_2_ without significantly reducing the hydrogen storage capacity [14,29,30,31,32]. Liang et al. [33] reported improved hydrogen storage performance of MgH_2_ with 5 wt % transition metals (Ti, V, Mn, Fe, and Ni), in which 5 wt % hydrogen was released from V-modified composite within 200 s at 300 °C. In recent years, the hydrogen storage system on Mg doped with Ti had attracted wide attention from researchers [34]. Shao et al. [35] synthesized a MgH_2_/0.1TiH_2_ composite by ball milling Mg and Ti powders under initial hydrogen pressure of 30 MPa and found that its dehydrogenation temperature was 100 °C lower than that of pure MgH_2_. Patelli et al. [36] used Mg–Ti vapors to synthesize Mg–Ti–H nanoparticles by reaction gas-phase condensation in He/H_2_ atmosphere of 2.6 mbar and the reversible absorption of hydrogen with MgH_2_–Mg phase transitions were achieved in a remarkably low 65–150 °C temperature range. Ma et al. [37] reported that the Mg–TiH_1.971_–TiH–Nb nanocomposite could absorb 5.6 wt % H_2_ within 5 min at 298 °C and 4.5 wt % H_2_ within 5 min at 250 °C, while pure Mg could absorb only 4 and 1.5 wt % H_2_ at the same temperature. 

Noteworthy, all the Ti in the above investigations were charged to Ti hydrides in the cycling, inspiring the researches add Ti hydrides directly to improve the hydrogen storage properties of MgH_2_. Bhatnagar et al. [18] reported that the MgH_2_–TiH_2_ system showed a lower enthalpy of desorption as compared to ball-milled MgH_2_ by about 7 kJ/mol. Jangir et al. [38] observed the onset temperature of MgH_2_ modified with TiH_2_ was 100 °C lower than that of as-milled MgH_2_ and the hydrogen release activation energy was decreased from –137 (MgH_2_) to –78 kJ/mol. Choi et al. [39] found the 7MgH_2_/TiH_2_ composite desorbed hydrogen at 126 °C, about 255 °C lower than that of pure MgH_2_ and the hydrogen storage performance of the composite system remained basically unchanged after five cycles.

Though many previous studies had explored the effects of the Ti hydrides on the hydrogen storage properties of MgH_2_, the size effect of Ti hydrides on its catalytic efficiency still needs to be pointed out. In this study, TiH_1.971_ with different particle sizes are prepared via a wet chemical ball milling method and the as-milled TiH_1.971_ nanoparticles are added to MgH_2_ to improve its hydrogen storage performance. XRD and TEM techniques are adopted to analyze the microstructure of the samples, and the hydrogen storage data are measured by DSC and a Sievert’s type apparatus. In addition, the possible catalytic mechanism is also discussed.

## 2. Experimental

All the primary materials were commercially available and used as-received without further purification, which included Mg (99.99%, Aladdin, 100-200 mesh, Sinopharm Chemical Reagent Co, Shanghai, China), TiH_1.971_ (99.99%, Alfa Aesar China Chemical Co, Shanghai, China), oleic acid (90%, Sinopharm Chemical Reagent Co, Shanghai, China), n-heptane (98.5%, Sinopharm Chemical Reagent Co, Shanghai, China), and oleamine (98%, Sinopharm Chemical Reagent Co, Shanghai, China). The Mikrouna glove box and Ball mill (QM-3SP4, Nanjing, China) were the instruments we used [40]. 

The MgH_2_ powders were synthesized by heat treatment and mechanical ball milling. First, the Mg powders were heated by a Sieverts-type volumetric apparatus at 380 °C and hydrogen pressure of 65~70 bar. Then, the samples were ball-milled at a speed of 450 rpm for 5 h in a planetary ball mill system (QM-3SP4, Nanjing, China). The ball-to-powder ratio (by weight) was 40:1. Subsequently, repeating the above two steps. Finally, the products of MgH_2_ were obtained after absorbing hydrogen under 380 °C and hydrogen pressure of 65~70 bar.

The nano-TiH_1.971_ powders were prepared by wet chemical ball milling. Specifically, the received TiH_1.971_ was mixed with oleic acid, oleamine, and heptane at a volume ratio of 1:0.33-1:10-1:20. After that, the mixtures were milled at a speed of 400 rpm for 30~60 h. The ball-to-powder weight ratio was 60:1. Then, the slurry was cleaned, stood, and centrifuged. Finally, TiH_1.971_ powders were obtained after being dried under vacuum condition for 10 h. 

MgH_2_–TiH_1.971_ composites were prepared by a mechanical ball milling method. Primarily, the prepared 5 wt % TiH_1.971_ with different milling time was introduced into as-synthesized MgH_2_ at 400 rpm for 4 h. During this process, the ball-to-powder weight ratio was 40:1. The samples were labeled as MgH_2_ + 5 wt %-*x*-TiH_1.971_, where *x* stood for the milling time (*x* = 30 h, 45 h, and 60 h, which were marked as *a*, *b,* and *c* in the following, respectively). On the basis, different amounts of TiH_1.971_ in the same milling time were added to MgH_2_ in the same way, marked as MgH_2_+ *y* wt % TiH_1.971_ (*y* = 1, 3, 5, and 7).

The de-/hydrogenation reaction in this paper occurred in a high pressure of gas absorption and desorption tester designed and assembled independently in the laboratory. The data of the sample’s temperature and pressure changes were recorded by the computer. The de-/hydrogenation performances of the samples (approximately 160~220 mg) were tested under the condition of isothermal and non-isothermal. In addition, the dehydrogenation and hydrogenation tests should be measured under the vacuum and the hydrogen pressure of 3 MPa, respectively. During the X-rays diffractometer (XRD) measurement, the samples were sealed in a custom-designed container and the data were collected in a 2*θ* range of 20–80° with 5°/min step increments in an X’Pert Pro X-ray diffractometer (PAN alytical, Royal Dutch Philips Electronics Ltd, Amsterdam, Netherlands) with Cu Kα radiation at 40 kV, 40 mA. The morphology and element distribution of the samples were characterized by transmission electron microscopy (TEM, Tecnai G2 F20 S-TWIN, FEI, Hillsboro, OR, USA) [40]. The differential scanning calorimetry (DSC, Netzsch STA 449F3, NETZSCH-GerätebauGmbH, Selb, Germany) tests for MgH_2_ and MgH_2_/TiH_1.971_ systems were conducted on an analyzer model from room temperature to 450 °C at different heating rates (5, 8, 10, 12 °C/min) with flowing argon (99.999%, 50 mL/min). 

## 3. Results and discussion

### 3.1. Characterization of Nano-TiH_1.971_

The elemental composition, structure, and morphology of the as-milled TiH_1.971_ were analyzed by XRD and TEM and the results are exhibited in Figure 1. Clearly, the particle size of TiH_1.971_ was decreased with increasing ball milling time, shown in Figure 1a–c. The particle size of TiH_1.971_ was 300 nm after ball milling for 30 h. When the milling time was increased to 45 h, the particle size decreased slightly to about 250 nm. Further increasing the milling time to 60 h, most of the TiH_1.971_ particles had the size of 150 nm. In addition, even with a long milling time, TiH_1.971_ phase (PDF#07-0370) still dominated the XRD pattern in Figure 1d. Besides, the average crystallite size of TiH_1.971_ was evaluated via the Debye–Scherrer equation [41] to be around 9.2 nm for the lattice planes (100), (200), (220), and (311), indicating the TiH_1.971_ particles were composed of nanocrystals. Based on the TEM and XRD results, it could be seen that TiH_1.971_ nanoparticles could be successfully synthesized by our method.

### 3.2. Catalytic Effect of TiH_1.971_ on Dehydrogenation of MgH_2_


To reveal the catalytic effectiveness of prepared TiH_1.971_ for desorption process, pure MgH_2_ and MgH_2_-TiH_1.971_ nanocomposites were subjected to non-isothermal and isothermal dehydrogenation tests. Figure 2a presents the non-isothermal dehydrogenation curves of pure MgH_2_ and MgH_2_ + 5 wt %-*x*-TiH_1.971_ composites. Compared with additive-free MgH_2_, both the onset and terminal dehydrogenation temperatures of TiH_1.971_ modified MgH_2_ systems were significantly reduced. Specifically, the dehydrogenation process of pristine MgH_2_ occurred at 312 °C~400 °C. Usually, this period of operation temperature was too high to satisfy the needs of practical application. After being doped with TiH_1.971_, the initial dehydrogenation temperature of MgH_2_ + 5 wt %-*c*-TiH_1.971_ composite decreased to about 175 °C, which was 137 °C lower than that of additive-free MgH_2_. Figure 2b shows the dehydrogenation curves under the isothermal mode at 300 °C. As shown in the picture, all the MgH_2_ + 5 wt %-*x*-TiH_1.971_ composites completed the dehydrogenation process within 10 min while the pure MgH_2_ could hardly release any hydrogen at the same condition. Moreover, the hydrogen desorption temperature clearly reduced because of the decrease of TiH_1.971_ particle size, where the MgH_2_ + 5 wt %-*c*-TiH_1.971_ composite could release hydrogen at 175 °C and 7.01 wt % H_2_ could be desorbed within 500 s at 300 °C. Besides the factor of particle size on the catalytic effect, the adding amount was also taken into consideration. Figure 2c displays the non-isothermal desorption curves of MgH_2_ and MgH_2_ + *y* wt %-*c*-TiH_1.971_ (*y* = 1, 3, 5, and 7) composites. Obviously, the dehydrogenation property of the composite could be improved immediately after adding only 1 wt % TiH_1.971_. The initial dehydrogenation temperature of MgH_2_ + 1 wt %-*c*-TiH_1.971_ decreased to about 200 °C, which was 110 °C lower than that of pure MgH_2_. With the increasing amount of TiH_1.971_, the dehydrogenation kinetics accelerated obviously. In addition, the dehydrogenation temperature for MgH_2_ + 7 wt %-*c*-TiH_1.971_ was almost the same as that of MgH_2_ + 5 wt %-*c*-TiH_1.971_ while the dehydrogenation capacity was decreased. Hence, the MgH_2_ + 5 wt %-*c*-TiH_1.971_ composite was chosen for further investigation because of its superior dehydrogenation kinetics and a relative high capacity.

Figure 3 shows the isothermal dehydrogenation curves of pure MgH_2_ and MgH_2_ + 5 wt %-*c*-TiH_1.971_ at different temperatures. The pure MgH_2_ desorbed only 0.34 wt % hydrogen within 10 min at 325 °C. However, the MgH_2_ + 5 wt %-*c*-TiH_1.971_ composite could discharge 7.0 wt % hydrogen within 10 min at 300 °C. Even at lower temperature of 250 °C, the 5 wt %-TiH_1.971_-containing sample still could liberate approximately 4.9 wt % H_2_ within 50 min. Figure 3b,d normalizes the hydrogen absorption curves by dividing the experimental hydrogen release from the ideal hydrogen containing (7.6 wt %). If the obtained result was closed to 1, it indicated that samples reached the saturated hydrogen absorption amount and the experiment data were effective and reliable [42]. 

The mechanism of hydrogen evolution was further researched by the kinetic solid-state reaction formula. Nucleation and growth, geometric shrinkage, diffusion mobility, and reaction sequence were applied to describe the experimental results, and the control steps of dehydrogenation rate were determined.

In general, dynamics equations for most solid-phase reactions could be described as [43]:f (α) = k t(1)
where α was the progress of solid-state reaction when the reaction time was t, k was the reaction rate constant. Sharp et al. [44] improved the formula as (2) to more simply and quickly select the most suitable kinetic model among the nine kinetic model characterization equations [44]:f (α) =A t/t0.5(2)
where A is the computable constant related to dynamic models, and t_0.5_ was the time value when α was 0.5. In brief, a linear relation graph was produced by drawing experiments of (t/t_0.5_)theo versus the theoretical values (t/t_0.5_)exp and the fitting linear slope value of the acceptable model should be close to 1. The representation equations of nine different dynamic models are shown in Table 1 [45].

The suitable kinetics reaction models for pure MgH_2_ and MgH_2_ + 5 wt %-*c*-TiH_1.971_ systems were applied to the isothermal dehydrogenation tests. Figure 4a shows the relationship of (t/t_0.5_)theo versus (t/t_0.5_)exp for pure MgH_2_ at 375 °C and the fitted linear slopes of the nine dynamics models are also listed in the picture. The A2 model had a best linear relationship because of its slope was 0.9992, which was very close to 1. Thus the nucleation and growth model of A2 (Avarami-Erofe’ev) fitted well with the kinetic data of synthesized molecule of MgH_2_. The kinetic model changed from A2 to R2 (see in Figure 4c) after adding TiH_1.971_ nanoparticles, indicating the isothermal dehydrogenation process of MgH_2_ + 5 wt %-*c*-TiH_1.971_ composite was controlled by the two-dimensional phase boundary model. Moreover, isothermal dehydrogenation curves of pure MgH_2_ and MgH_2_ + 5 wt%-*c*-TiH_1.971_ composites at other temperatures were all well interpreted by A2 and R2 models (Figure 4b,d), demonstrating these kinetic models could truly explain the dehydrogenation process. As Ti has the medium electronegativity between Mg and H (Ti (1.5), Mg (1) and H_2_ (2)), Ti ions are easier to gain or lose electrons (e^–^) than Mg ions or H^-^ ions. In addition, the ball milling process created a favorable contact between the TiH_1.97_ and MgH_2_. Hence, TiH_1.971_ could act as an intermediate carrier during the electron transferring between Mg^2+^ and H^–^. Besides, the particle size of the composite after ball milling was in the range of nanometers, [17] which would of course reduce the hydrogen diffusion distance. It was also proved that the nucleation and crystal growth process were not controlled by intraparticle diffusion but via the surface conversion of MgH_2_ [46]. In our case, the abundance of polymorphic states of MgH_2_ and their slow interphase boundary migration might affect the dehydrogenation kinetics, [42] making Mg–MgH_2_ phase boundary movement the rate limiting step of the isothermal decomposition process in the MgH_2_ + 5 wt %-*c*-TiH_1.971_ composite under current experimental conditions.

In order to further study the improvement on the hydrogen desorption behaviors of TiH_1.971_ doped into MgH_2_, the activation energies (E_a_) of dehydrogenation for MgH_2_ and MgH_2_ + 5 wt %-*c*-TiH_1.971_ were calculated by the Kissinger formula. The Kissinger’s equation could be written as [17]:ln (C/T_P_^2^) = - (E_a_/(RT_P_)) + A(3)
where C was the heating rate, T_P_ was the peak temperature at the corresponding hydrogen production rate, R was the gas constant, A was temperature-independent constant.

DSC curves with various heating rates (5, 8, 10, and 12 K/min) for MgH_2_ and MgH_2_ + 5 wt %-*c*-TiH_1.971_ composites are presented in Figure 5a. Clearly, the peak temperatures of the MgH_2_ + 5 wt %-*c*-TiH_1.971_ composites were significantly lower than that of pure MgH_2_ at every heating rate. Figure 5b reveals that the activation energies were 83 ± 7 kJ/mol for MgH_2_ + 5 wt %-*c*-TiH_1.971_ and 155 ± 16 kJ/mol for pure MgH_2_. Thus, the E_a_ value of MgH_2_ + 5 wt %-*c*-TiH_1.971_ was 46.45% lower than that of pristine MgH_2_, which was also competitive with reported ZrMn_2_, Ni/CMK-3, TiO_2_@C, Ta_2_O_5_, Li_2_TiO_3_ modified MgH_2_ systems [14,29,30,31,32].

### 3.3. Catalytic Effect of TiH_1.971_ on Hydrogenation of MgH_2_


In order to further study the catalytic effect of TiH_1.971_ on hydrogen absorption, the hydrogenation kinetics of MgH_2_ + 5 wt %-*c*-TiH_1.971_ composite and pure MgH_2_ were measured. The non-isothermal hydrogenation data graphs of MgH_2_ with and without TiH_1.971_ are showed in Figure 6a. It is clear that the onset rehydrogenation temperature was significantly reduced after the introduction of TiH_1.971_. Especially, the dehydrogenated MgH_2_ + 5 wt %-*c*-TiH_1.971_ samples could absorb hydrogen even at room temperature (23 °C), which was 125 °C lower than that of pure MgH_2_. Figure 6b presents the isothermal hydrogenation curves for pure MgH_2_. When the temperature was heated up to 250 °C, the dehydrogenated MgH_2_ sample could be completely hydrogenated and approximately 4.5 wt % H_2_ was charged within 1 h at 210 °C. After TiH_1.971_ was doped, the hydrogen absorption performance of MgH_2_ was obviously enhanced (Figure 6c). The dehydrogenated MgH_2_ + 5 wt %-*c*-TiH_1.971_ sample took up 4.4 wt % H_2_ within 1 h even at low temperature of 60 °C. When the hydrogenation reaction was performed at 125 °C, 3.8 wt % H_2_ could be absorbed in only 4 min. 

In addition, the apparent activation energies (E_a_) of hydrogen absorption for MgH_2_ and MgH_2_ + 5 wt %-*c*-TiH_1.971_ were also calculated. The Johnson-Mehl-Avrami-Kolmogorov (JMAK) model, studied by Avrami, could be used to fit the isothermal hydrogen adsorption kinetics curves. The interface velocity of MgH_2_ formation was assumed as constant. The linear equation was as follows [47]:ln [-ln (1-α)] = η ln (k)+ η ln (t),(4)
where α was the hydrogenation reaction degree of MgH_2_ at the dimension of reaction time of t; the transformation rate constant was k; and η was the growth of MgH_2_. Based on the above hydrogenation curves, Figure 6d,e presents the fitted curves for each temperature. It is evident that the fitting degree of each curve was in accordance with the values of R^2^, which were all over 0.98. Then the E_a_ values of hydrogenation were calculated from the Arrhenius equation [17]:k= A exp (Ea/RT),(5)
where A is the temperature-independent coefficient, T is the thermodynamic temperature. In Figure 6f, the hydrogenation E_a_ of MgH_2_ + 5 wt %-*c*-TiH_1.971_ sample was 49 ± 4 kJ/mol, much lower than that of the pure MgH_2_ (73 ± 3 kJ/mol), which played an important role in significantly improving the hydrogen absorption properties of MgH_2_.

### 3.4. Cycling Hydrogen Storage Properties of the MgH_2_-TiH_1.971_ Composites

Besides the hydrogen desorption and absorption performance, cycling performance is also highlighted in the present study. To verify the effect of TiH_1.971_ on reversibility, the cycling performance of MgH_2_ + 5 wt %-*c*-TiH_1.971_ composites were tested for 10 times, displayed in Figure 7. It is found that the catalyzed MgH_2_ could stably release 7.0 wt% H_2_, and the hydrogenation reaction could be quickly completed within 1 min. Previous research reported that it was difficult for the hydrogen molecules to enter into the clusters and lead to a poor cycling reversibility as MgH_2_ particles would gather into clusters during heating [48]. However, what can be seen from Figure 7 that all the cycles presented the fast hydrogen uptake kinetics and the hydrogen storage capacity decreased very slightly in this study. This might result from the dispersed TiH_1.971_ that stood in the way of sintering and growth of MgH_2_, which contribute to maintain a relatively stable cyclic reversibility.

### 3.5. Evolution of TiH_1.971_ during Cycling and Its Catalytic Mechanism

Figure 8 presents the XRD patterns of dehydrogenated and rehydrogenated MgH_2_/TiH_1.971_ samples, in which Mg or MgH_2_ were the main phase after desorption or absorption. It was worth noting that both the states showed the signal of TiH_1.971_, demonstrating the stability in our operating conditions, which was in agreement with a previous report that Ti hydride only decomposed over 420~575 °C [34]. The XRD results verified that TiH_1.971_ was stable and could be served as an active catalyst persistently in the hydrogen uptake.

Among many theories about the role of catalytic additives, there are two hypotheses: “hydrogen spillover” and “hydrogen gateway” [49,50,51]. “Hydrogen spillover” means a catalysis effect in which hydrogen molecules firstly dissociate on the surface of additive particles and then “spillover” to metal atoms to form a hydride. “Hydrogen gateway” is a phase transformation process in which hydrogen is absorbed by the additive particles to form the additive’s hydride phase, and then reacts with the metallic particles to form a metal hydride. In our case, TiH_1.971_ nanoparticles remained stable in the de/rehydrogenation process, agrees well with the theory of “hydrogen spillover” [52]. In detail, hydrogen molecules were free on the surface of TiH_1.971_ nanoparticles, and hydrogen atoms were easily transferred to the surface of Mg particles to form MgH_2_ during the hydrogenation process. Similarly, TiH_1.971_ nanoparticles also effortlessly took up hydrogen atoms from the MgH_2_ matrix to form hydrogen molecules in the dehydrogenation reaction [53]. In this way, the activation energies for de/rehydrogenation were reduced and the de/rehydrogenation kinetics were enhanced.

## 4. Conclusions

In a word, TiH_1.971_ nanoparticles with different particle size were successfully synthesized and showed excellent catalytic effects in improving the hydrogen storage performances of MgH_2_. The effects of particle size and concentration of TiH_1.971_ on hydrogen storage performance of MgH_2_ were studied and compared, and the optimum composite of MgH_2_ + 5 wt %-*c*-TiH_1.971_ was obtained. The MgH_2_ + 5 wt %-*c*-TiH_1.971_ composite system started to release hydrogen at 175 °C, which was 137 °C lower than the as-synthesized MgH_2_. In addition, a suitable dynamic model was applied by fitting the isothermal dehydrogenation curves. Besides, the dehydrogenation activation energy of MgH_2_ was decreased from 155 ± 16 kJ/mol to 83 ± 7 kJ/mol after doping with TiH_1.971_. For hydrogenation, the dehydrogenated MgH_2_ + 5 wt %-*c*-TiH_1.971_ sample could absorb 4.60 wt % hydrogen in 10 min at 125 °C under 3 MPa hydrogen pressure. In contrast, the pure MgH_2_ only absorbed 4.11 wt % hydrogen in the same time at higher temperature of 250 °C. The apparent hydrogenation activation energy of MgH_2_ + 5 wt %-*c*-TiH_1.971_ was 49 ± 4 kJ/mol, which was nearly 32.87% lower than that of pure MgH_2_ (73 ± 3 kJ/mol). Moreover, the MgH_2_ + 5 wt %-*c*-TiH_1.971_ composite showed superior cyclic stability. The direct use of TiH_1.971_ nanoparticles as catalyst will be helpful for understanding the design and preparation of more efficient materials for hydrogen storage in the future.

## Figures and Tables

**Figure 1 nanomaterials-09-01370-f001:**
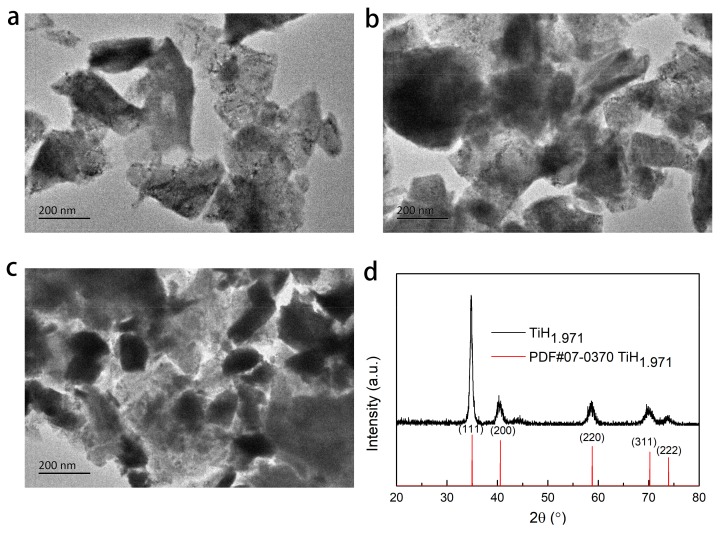
TEM of TiH_1.971_ milled for 30 (**a**), 45 (**b**), 60 (**c**) hours and XRD pattern of TiH_1.971_ milled for 60 h (**d**).

**Figure 2 nanomaterials-09-01370-f002:**
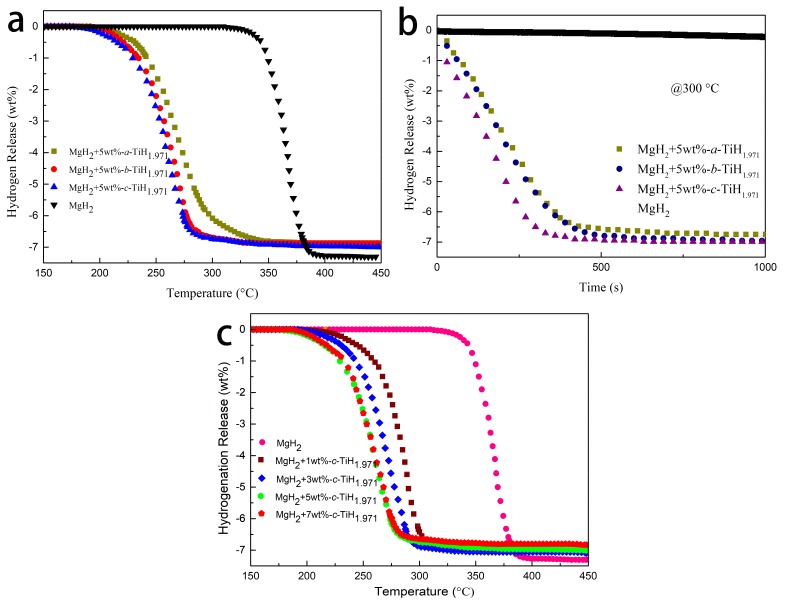
Non-isothermal desorption curves (**a**) of MgH_2_ and MgH_2_ + 5 wt%-*x*-TiH_1.971_ (*x* = *a*-30, *b*-45 and *c*-60) samples, isothermal desorption curves (**b**) of MgH_2_ + 5 wt%-*x*-TiH_1.971_ samples at 300 °C, non-isothermal desorption curves (**c**) of MgH_2_ and MgH_2_ + *y* wt%-*c*-TiH_1.971_ (*y* = 1, 3, 5, and 7) samples.

**Figure 3 nanomaterials-09-01370-f003:**
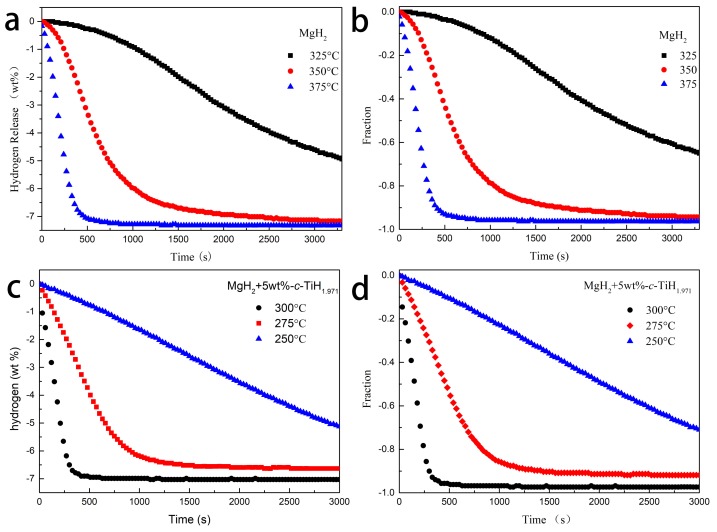
Isothermal dehydrogenation curves (**a**) and normalized isothermal dehydrogenation curves (**b**) from prepared MgH_2_ at different temperatures, isothermal dehydrogenation curves (**c**), and normalized isothermal dehydrogenation curves (**d**) from MgH_2_ + 5 wt%-*c*-TiH_1.971_ at different temperatures.

**Figure 4 nanomaterials-09-01370-f004:**
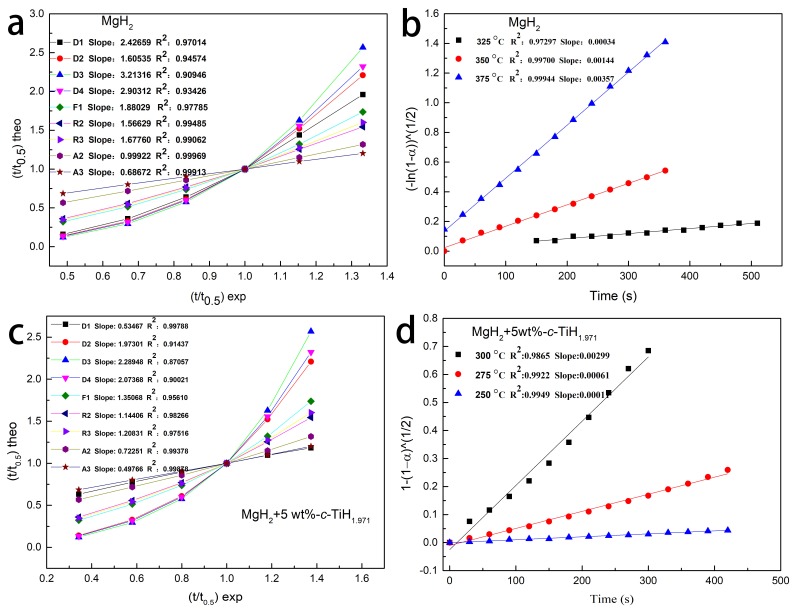
The (t/t_0.5_)theo vs. (t/t_0.5_)exp (**a**) for MgH_2_ at 375 °C using various kinetic models, the time dependence of the kinetic modeling equation (**b**) for MgH_2_ at different temperatures. The (t/t_0.5_)theo vs. (t/t_0.5_)exp (**c**) for MgH_2_ + 5 wt%-*c*-TiH_1.971_ at 300 °C using various kinetic models. The time dependence of the kinetic modeling equation (**d**) for MgH_2_ + 5 wt%-*c*-TiH_1.971_ at different temperatures.

**Figure 5 nanomaterials-09-01370-f005:**
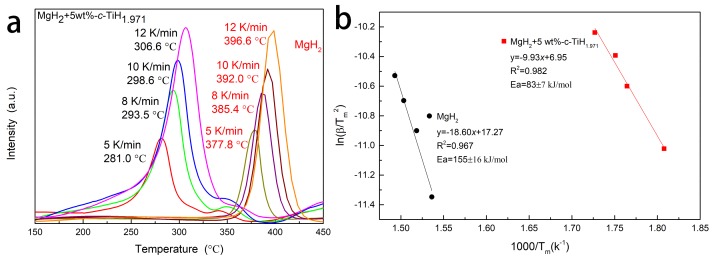
DSC curves of (**a**) prepared MgH_2_ and MgH_2_+5 wt%-*c*-TiH_1.971_ at various heating rates (5, 8, 10, and 12 °C/min) and estimations of the apparent active energies using the Kissinger method with the parameters obtained from DSC measurements (**b**).

**Figure 6 nanomaterials-09-01370-f006:**
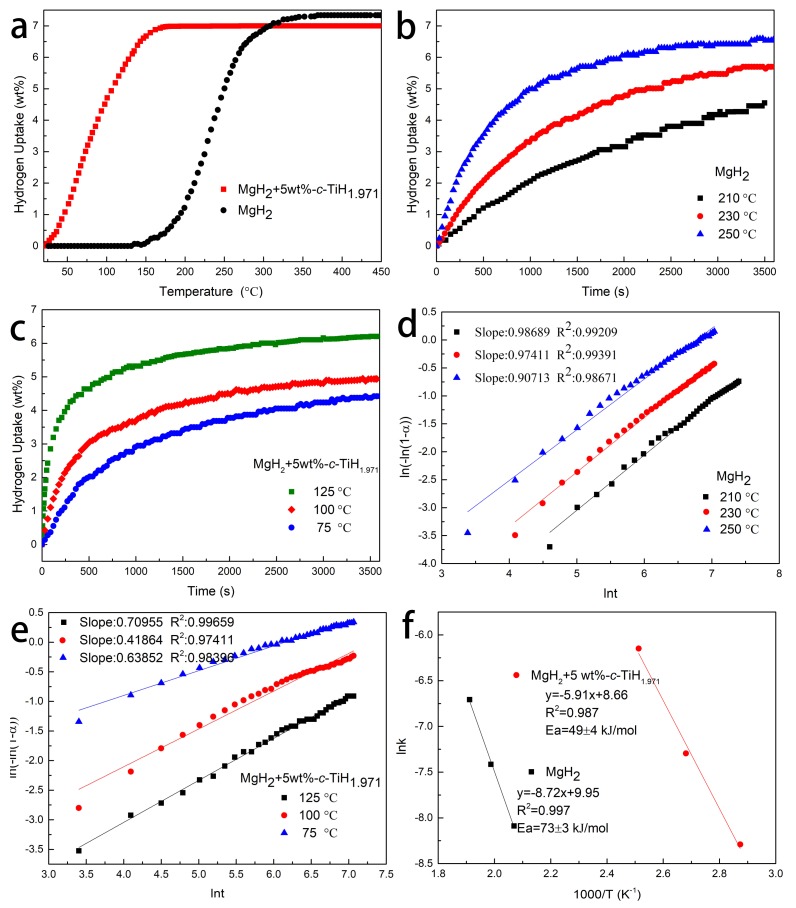
Non-isothermal hydrogenation curves (**a**) of the MgH_2_ and MgH_2_ + 5 wt%-*c*-TiH_1.971_ samples, isothermal hydrogenation curves of the prepared MgH_2_ (**b**), and MgH_2_ + 5 wt%-*c*-TiH_1.971_ (**c**) samples at different temperatures, isothermal hydrogenation JMAK curve plots of the prepared MgH_2_ (**d**) and the MgH_2_ + 5 wt%-*c*-TiH_1.971_ (**e**) samples, the fitted Arrhenius curve plots (**f**) of MgH_2_ and MgH_2_ + 5 wt%-*c*-TiH_1.971_ samples.

**Figure 7 nanomaterials-09-01370-f007:**
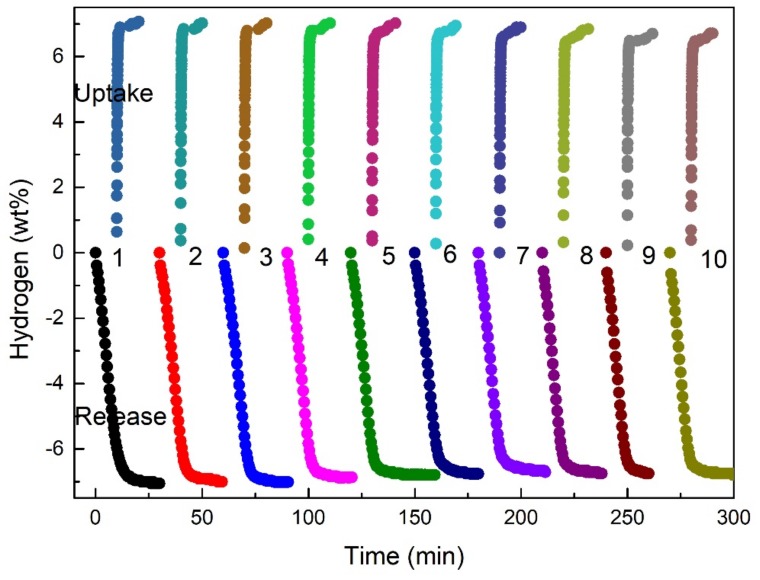
Cycling profiles of MgH_2_ + 5 wt%-*c*-TiH_1.971_ under dehydrogenation conditions of 300 °C, and hydrogenation conditions of 300 °C under 3 MPa H_2_.

**Figure 8 nanomaterials-09-01370-f008:**
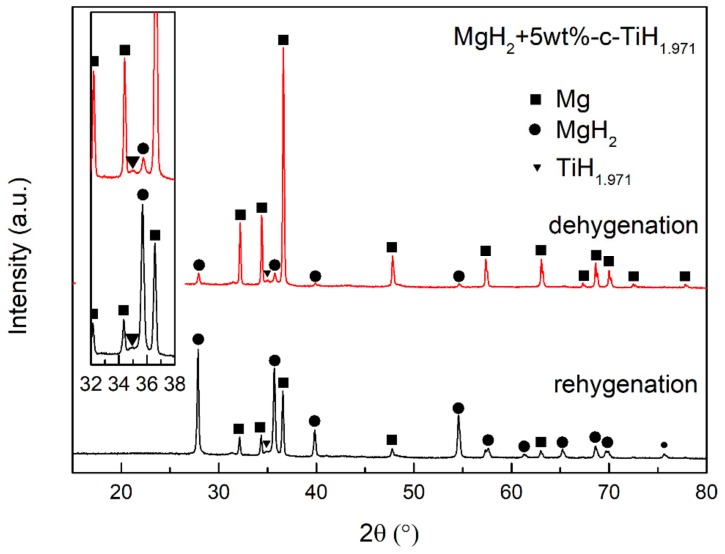
XRD patterns of dehydrogenated and rehydrogenated MgH_2_/TiH_1.971_ samples.

**Table 1 nanomaterials-09-01370-t001:** Different kinetic models for isothermal desorption.

Symbol	Model	Integral *f*(α) Form
D1	One-dimensional diffusion	α^2^
D2	Two-dimensional diffusion	α+(1-α)ln(1-α)
D3	Three-dimensional diffusion(Jander equation)	[1-(1-α)^1/3^]^2^
D4	Three-dimensional diffusion(Ginstling-Braunshtein equation)	(1-2α/3)-(1-α)^2/3^
F1	First-order reaction	-ln(1-α)
R2	Two-dimensional phase boundary	1-(1-α)^1/2^
R3	Three-dimensional phase boundary	1-(1-α)^1/3^
A2	Avarami-Erofe’ev	[-ln(1-α)]^1/2^
A3	Avarami-Erofe’ev	[-ln(1-α)]^1/3^

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
