# Peer review of "Catalytic Effect of Facile Synthesized TiH1.971 Nanoparticles on the Hydrogen Storage Properties of MgH2"

_nanomaterials, 2019, doi:10.3390/nano9101370_

Round 1

Reviewer 1 Report

Investigations on the kinetic behavior of MgH2 with TiH1.97 as additive were done. It was found that the additive improves the kinetic behavior of MgH2. Moreover, based on the gas-solid models fitting, the Ea for hydrogenation and dehydrogenation were calculated. It was reported that the presence of the additive lowers the hydrogenation and dehydrogenation Ea.

 I consider that the work can be published after the following points have been addressed:

 General comments:

 I would like to suggest going through the manuscript again carefully for clarity, syntax and correctness. The English should be improved. Please, replace the word “abate” for “decrease” or “reduce”. It is demanding to check the tenses throughout the paper.

 Abstract

 The abstract should be re-written according to the comments.

 Please, carefully check the English.

 Introduction

2.1       Please, add updated references for the first general sentence about the need of renewable energy sources.

2.2       References 2 and 3 and not representative of the second sentence. Please, replace it for:

Züttel, A.; Borgschulte, A.; Schlapbach, L. Hydrogen as a Future Energy Carrier; WILEY-VCH Verlag: Weinheim,

Germany, 2008.

 2.3       Page 1, sentence: Up to now, chemical hydrides, metal hydrides, metal-organic frameworks (MOFs) and adsorbents including carbon have been studied as hydrogen storage media. [6-10]

Please, try to refer to review works in which the state of art of this materials is thoroughly described. The referred works (6-10) are quite specific and as far as I am concerned not representative for the description of the state of art.

2.4       Page 1, sentence:

 Tao et al.[14] considered that the ideal desorption energy of MgH2 was 75 kJ/mol (Td=300 °C), and the ideal absorption energy was between 20 and 50 kJ/mol (Td=20-100 °C).  

This sentence is nor clear. What do you mean by “ideal desorption energy”? Would it be “theoretical desorption enthalpy” and dehydrogenation temperature of 300 ºC at 1 bar of H2? Then, the “the ideal absorption energy was between 20 and 50 kJ/mol (Td=20-100 °C)¨, Do you mean that the desorption enthalpy should be between 20 and 50 kJ/mol H2 in order to have a dehydrogenation temperature at 1 bar H2 between 20 and 100 ºC?

Please, modify this sentence.

2.5       Page 1 – 2: But the Mg-H bond was very stable and difficult to break,[14] leading to higher dehydrogenation temperature  (300~400 °C)[15-17] and slower kinetic performance (1 wt % H2/min, at 300 °C).[18]

It is true that the bond between Mg and H is quite strong and this is given by the thermodynamic stability, i.e. “enthalpy”. Please, try to re-write the sentence mentioned in 2.4 and this one.

2.6       Page 2: In order to overcome this problem, researchers worldwide tried oceans of modification technics to improve the hydrogen storage performance of MgH2, including nanostructuring, surface modification, alloying and doping with transition metals.[19-28]

Some strategies to improve the hydrogen storage properties of MgH2 are here described. However, one of the main approach used to improve MgH2 is nor mentioned. This method is “destabilization”. Please, include this strategy which addresses the change of thermodynamic stability of MgH2 by the combination for instance with another hydride and formation of compounds upon dehydrogenation.

In general, the introduction does not provide a good scope about the field of research. There are several concept that are not properly described.

Experimental

3.1       The experimental is a little bit messy. Please, try to structure this section in order to provide the reader better comprehension. For instance, at the beginning of this section, it is mentioned that a Sieverts device is used. Then, the synthesis of the materials is described. In the last paragraph, the details of the Sieverts measurements are given. This sounds untidy. 

Results and discussion

 4.1       Page 8 and 9, section 3.2: Please, include de band error at the time to express the activation energies. For instance, the Ea in Fig. 5 is expressed as 154.6 ± 16.3 kJ/mol. This should be expressed as 155±16 kJ/mol. The same approach should be used for the other values. Please, based on this suggestions, the expression of the Ea numbers in the manuscript should be changed.

4.2       Page 9 and 10, section 3.3: Why did not you use the reduced time method to determine the rate limiting step upon hydrogenation? It should be advisable to try to measure an additional curve in order to have at least 4 points to calculate the hydrogenation Ea.

4.3       Page 11 and 12, section 3.5: This section does not provide any insight about the “catalytic mechanism”. It is just a description about the stability of the TiH1.97 but it does not provide any detail about the effect of the additive on the observe improved kinetic behavior. What would it be effect of the additive? Would it be an effect on the recombination of the hydrogen molecule? Would the additive be on the grain boundary of Mg, at the interphase of MgH2/Mg? Please, try to discuss these aspects based on the results and if possible, try to do more TEM observations, HR-TEM observations, in order to verify where the additive´s particle are.

Conclusions

 5.1       The conclusions should be re-written according to the comments.

References

6.1       The references should be more representative in order to better describe the scope of the work.

Author Response

The amendment and rebuttal list for the manuscript of

Nanomaterials-584373

The main corrections in the paper and the responds to the reviewer’s comments are as flowing:

Reviewer #1:

Investigations on the kinetic behavior of MgH2 with TiH1.97 as additive were done. It was found that the additive improves the kinetic behavior of MgH2. Moreover, based on the gas-solid models fitting, the Ea for hydrogenation and dehydrogenation were calculated. It was reported that the presence of the additive lowers the hydrogenation and dehydrogenation Ea. I consider that the work can be published after the following points have been addressed:

1) I would like to suggest going through the manuscript again carefully for clarity, syntax and correctness. The English should be improved. Please, replace the word “abate” for “decrease” or “reduce”. It is demanding to check the tenses throughout the paper.

Author reply:

Thanks for your comment. The word “abate” has been replace for “decrease” in the revised manuscript. The language of the entire article has been carefully checked again.

2) The abstract should be re-written according to the comments. Please, carefully check the English.

Author reply:

Thanks for the reviewer’s comment. The abstract has been re-written according to the comments and the language has been checked in the revised manuscript.

3) Please, add updated references for the first general sentence about the need of renewable energy sources.

Author reply:

Thanks for the reviewer’s advice. The updated references for the need of renewable energy sources has been added in the revised manuscript.

4) References 2 and 3 and not representative of the second sentence. Please, replace it for: Züttel, A.; Borgschulte, A.; Schlapbach, L. Hydrogen as a Future Energy Carrier; WILEY-VCH Verlag: Weinheim, Germany, 2008.

Author reply:

Thanks for the reviewer pointing out this question. The original references has been deleted, and the articles mentioned by the reviewer have been added in the revised manuscript.

5) Page 1, sentence: Up to now, chemical hydrides, metal hydrides, metal-organic frameworks (MOFs) and adsorbents including carbon have been studied as hydrogen storage media. [6-10] Please, try to refer to review works in which the state of art of this materials is thoroughly described. The referred works (6-10) are quite specific and as far as I am concerned not representative for the description of the state of art.

Author reply:

Thanks for the reviewer’s good advice. The relevant papers which thoroughly describe the state of art has been cited in the revised manuscript.

6) Page 1, sentence: Tao et al.[14] considered that the ideal desorption energy of MgH2 was 75 kJ/mol (Td=300 °C), and the ideal absorption energy was between 20 and 50 kJ/mol (Td=20-100 °C).

This sentence is not clear. What do you mean by “ideal desorption energy”? Would it be “theoretical desorption enthalpy” and dehydrogenation temperature of 300 ºC at 1 bar of H2? Then, the “the ideal absorption energy was between 20 and 50 kJ/mol (Td=20-100 °C)¨, Do you mean that the desorption enthalpy should be between 20 and 50 kJ/mol H2 in order to have a dehydrogenation temperature at 1 bar H2 between 20 and 100 ºC?

Please, modify this sentence.

Author reply:

Thanks for the reviewer pointing out this question. This sentence is the language in the reference. After reading the original literature again, we found that the reviewer's understanding is correct. In order to avoid confusion, we removed this sentence in the revised manuscript.

7) Page 1 – 2: But the Mg-H bond was very stable and difficult to break,[14] leading to higher dehydrogenation temperature (300~400 °C)[15-17] and slower kinetic performance (1 wt % H2/min, at 300 °C).[18] It is true that the bond between Mg and H is quite strong and this is given by the thermodynamic stability, i.e. “enthalpy”. Please, try to re-write the sentence mentioned in 2.4 and this one.

Author reply:

Thanks for the reviewer’s kind suggestion. We have re-written this sentence in the revised manuscript.

8) Page 2: In order to overcome this problem, researchers worldwide tried oceans of modification technics to improve the hydrogen storage performance of MgH2, including nanostructuring, surface modification, alloying and doping with transition metals.[19-28] Some strategies to improve the hydrogen storage properties of MgH2 are here described. However, one of the main approach used to improve MgH2 is nor mentioned. This method is “destabilization”. Please, include this strategy which addresses the change of thermodynamic stability of MgH2 by the combination for instance with another hydride and formation of compounds upon dehydrogenation. In general, the introduction does not provide a good scope about the field of research. There are several concept that are not properly described.

Author reply:

Thanks for the kind advise from the referee. The method of “destabilization” has been introduced in the revised manuscript. Besides, the introduction has been modified and some concepts have been described more clearly.

9) The experimental is a little bit messy. Please, try to structure this section in order to provide the reader better comprehension. For instance, at the beginning of this section, it is mentioned that a Sieverts device is used. Then, the synthesis of the materials is described. In the last paragraph, the details of the Sieverts measurements are given. This sounds untidy.

Author reply:

Thanks for the reviewer pointing out this question. We are sorry that we did not introduce the experiment clearly. We had reorganized the language of the experimental part in the revised manuscript.

10) Page 8 and 9, section 3.2: Please, include de band error at the time to express the activation energies. For instance, the Ea in Fig. 5 is expressed as 154.6 ± 16.3 kJ/mol. This should be expressed as 155±16 kJ/mol. The same approach should be used for the other values. Please, based on this suggestions, the expression of the Ea numbers in the manuscript should be changed.

Author reply:

Thanks for the good advice. We have changed the expression of Ea numbers in the revised manuscript.

11) Page 9 and 10, section 3.3: Why did not you use the reduced time method to determine the rate limiting step upon hydrogenation? It should be advisable to try to measure an additional curve in order to have at least 4 points to calculate the hydrogenation Ea.

Author reply:

Thanks for your comments. We don't know enough about the reduced time method mentioned by the reviewer. In the future study, we will focus on this method. Due to time constraints, this article will not discuss it in detail for the time being. Based on the curve fitted by the three points, we find that the deviation value (R2) is greater than 0.98. This means that these three points can already determine the slope of the fitted line and calculate the activation energy.

12) Page 11 and 12, section 3.5: This section does not provide any insight about the “catalytic mechanism”. It is just a description about the stability of the TiH1.971 but it does not provide any detail about the effect of the additive on the observe improved kinetic behavior. What would it be effect of the additive? Would it be an effect on the recombination of the hydrogen molecule? Would the additive be on the grain boundary of Mg, at the interphase of MgH2/Mg? Please, try to discuss these aspects based on the results and if possible, try to do more TEM observations, HR-TEM observations, in order to verify where the additive´s particle are.

Author reply:

Thanks for your comments and kind suggestion. By reading many articles, we thought that the theory of the TiH1.971 effects could be summarized as a type of “hydrogen spillover”, which suggested that the hydrogen molecules dissociated on the surfaces of the TiH1.971 nanoparticles and then the hydrogen atoms easily transferred to the surface of Mg particles to generate MgH2 during the hydrogenation process. Similarly, TiH1.971 nanoparticles also effortlessly took up hydrogen atoms from the MgH­2 matrix to form hydrogen molecules in the dehydrogenation reaction. As a result, the de-/rehydrogenation kinetics of MgH2 was greatly improved after being catalyzed by TiH1.971 nanoparticles. Such a description for “catalytic mechanism” had been added in the revised manuscript.

13) The conclusions should be re-written according to the comments.

Author reply:

Thanks for your comments. The conclusions have been re-written in revised manuscript.

14) The references should be more representative in order to better describe the scope of the work.

Author reply:

Thanks for your kind suggestion. More representative articles have been added and cited in the revised manuscript.

We appreciate for reviewer’s warm work earnestly, and hope that the correction will meet with approval.

Once again, thank you very much for your comments and suggestions.

Reviewer 2 Report

The paper provides a description of thorough investigation concerning the synthesis of MgH2 doped with TiH1.971 and study on its hydrogen storage performance. The conclusions are quite interesting and the explanation and ideas presented in the paper may be helpful for understanding the design and preparation of more efficient materials for hydrogen storage.
This quite important paper is probably publishable, however, there is a number of issues to be addressed:

1. The title: it states that the authors propose a facile method of synthesis of catalyst TiH1.971 and its effect on the hydrogen sorption kinetics. The first question is: do we really deal with sorption (i.e. physical process) or rather hydrogenation and dehyrogenation as chemical reactions? In this regard, the nomenclature used is inconsistent. The second question: in the text the authors claim that they study the influence of the size of nanocrystallites on the de-/hydrogenation. This is quite important information and should be emphasized in the abstract, introduction, experimental section and so on. If not, the novelty of the presented procedure is not convincing, because in the introduction we read that very similar systems hed been studied while nanocrystallite sizes were not taken into consideration.
2. The authors present some results of model calculations and propose how to describe the process in question using mathematical equations. However, I would like to ask the authors to add at least a few sentences of comments on theoretical explanation why the proposed system works the way it was observed. Why the system is so active and and effective? What is the mechanism of the process studied? How does the catalyst really work? From the text we got to know that e.g. the activation energies had changed. But why is it so? We got some pieces of information on page 7 (changes of models and so on, but what is the particular meaning of the scientific terms mentioned there e.g. two-dimensional phase boundary; what is the exact role of the components of the system? what is the active site of the catalyst and so on.
3. Page 1: Td = 300oC, but then we read Td = 20-100oC. I guess this should be Ta (from absorption).
4. In some places 'oC' (for temperature) are used and in other data we see 'K'. Can the authors convert the values and enter the same unit everywhere?
5. The experimental section:
- the second paragraph (MgH2 synthesis): experimental equipment should be described in more detail (scheme of the experimental setup?). Are the authors sure that the obtained samples were always the same?
- the fifth paragraph: 'the dates were collected' - unclear; 40 Kelvin Volt?
There is no description to the experimental setup for de-/hydrogenation reaction. Most information we read in the result and discussion section (a little bit too late).
6. Results and discussion section:
- the first paragraph (point 3.1.): XRD method may not 'see' impurities due to a small amount or non-crystalline form (so this is not the proof demonstrating the purity of the catalyst); what about using XRD method instead of TEM to determine mean size of nanoparticles in the samples? TEM may not may not distinguish agglomerates from single nanoparticles.
- the second paragraph (point 3.2.): the reader may be surprised finding out that the effect of catalyst concentration was also studied (there was nothing about it in the abstract and experimental section); the influence of a diameter is hardly visible in Fig. 2a and b; can you change the X-axis scale? what was the diameter after the reactions? (I think this was the real diameter of the nanoparticles - not before the temperature processes; especially when you mentioned crystal growth process e.g. on Page 7).
7. Page 5: 'the date of the experiment was reliable' - unclear.
8. Page 6, Fig. 3: why do we have different temperatures for both systems (with and without catalyst) here? These two systems should be investigated in the same ranges of all parameters (the same with Fig. 4).
9. Page 6:
- the first paragraph: this should be mentioned in the abstract, introduction (to emphasize the novelty of the paper) and conclusions.
- next: (t/t0)exp to (t/t0)theo relation is mentioned while in figures 4 we see the opposite relation.
- model equations and table 1: at what assumptions can we describe chemical reactions using kinetic models for isothermal desorption? Is the adsorption (or desorption) the rate limiting step of the studied reactions?
10. Page 7: 'The suitable kinetics reaction models for pure MgH2 and MgH2+5 wt%-c-TiH1.971 systems were discovered' - who discovered? The authors?
11. Page 7: there is not comment about the Fig. 4b.
12. Page 8: ZrMn2, Ni/CMK-3, TiO2@C, Ta2O5, Li2TiO3 modified MgH2 systems not mentioned in the introduction. Why?
13. Conclusions: 'the best optimum' - the best usually means somehow optimum. 'nearly half lower than' - when we compare ca. 50 kJ/mol and ca. 75 kJ/mol, I would say the change (decrease of 75 to 50) is ca. 30% (increase of 50 to 75 would be 50%). Am I right?
16. English language - I am NOT a specialist (I'm not an English teacher), however, I encountered many linguistic errors in the text. As one example, see the sentence of the introducion: 'Energy is one of the most important motive force for national economic development, however, the increasing content of pollutants such as carbon dioxide in the atmosphere WERE(?) caused by high consumption of fossil fuels forced a(?) urgent demand for cleaner and sustainable energy resources.'. Therefore, I strongly recommend a careful reading of the text by an English native speaker to make sure that everything is OK. Without this it is very difficult to clearly assess the quality of the work (ideas, results, conclusions are probably publishable though).

To conclude, perhaps, it would be enough only to significantly re-edit the text. This is probably only a matter of better presentation and description of current results.

Author Response

The amendment and rebuttal list for the manuscript of

Nanomaterials-584373

The main corrections in the paper and the responds to the reviewer’s comments are as flowing:

Reviewer #2:

The paper provides a description of thorough investigation concerning the synthesis of MgH2 doped with TiH1.971 and study on its hydrogen storage performance. The conclusions are quite interesting and the explanation and ideas presented in the paper may be helpful for understanding the design and preparation of more efficient materials for hydrogen storage. This quite important paper is probably publishable, however, there is a number of issues to be addressed:

1) The title: it states that the authors propose a facile method of synthesis of catalyst TiH1.971 and its effect on the hydrogen sorption kinetics. The first question is: do we really deal with sorption (i.e. physical process) or rather hydrogenation and dehydrogenation as chemical reactions? In this regard, the nomenclature used is inconsistent. The second question: in the text the authors claim that they study the influence of the size of nanocrystallites on the de-/hydrogenation. This is quite important information and should be emphasized in the abstract, introduction, experimental section and so on. If not, the novelty of the presented procedure is not convincing, because in the introduction we read that very similar systems had been studied while nanocrystallite sizes were not taken into consideration.

Author reply:

Thanks for the reviewer pointing out this question. For the first question, we are sorry that we do not have a good distinction between the concept of sorption and reaction. This paper is devoted to the study of hydrogen storage properties of MgH2 by chemical methods. So we have changed the title to “Catalytic effect of facile synthesized TiH1.971 nanoparticles on the hydrogen storage properties of MgH2” in the revised manuscript. For the second problem, the TiH1.971 studied in this paper is nanoparticle rather than the nanocrystal mentioned by the reviewer. According to the reviewer's suggestion, more information has been emphasized in the relevant sections.

2) The authors present some results of model calculations and propose how to describe the process in question using mathematical equations. However, I would like to ask the authors to add at least a few sentences of comments on theoretical explanation why the proposed system works the way it was observed. Why the system is so active and effective? What is the mechanism of the process studied? How does the catalyst really work? From the text we got to know that e.g. the activation energies had changed. But why is it so? We got some pieces of information on page 7 (changes of models and so on, but what is the particular meaning of the scientific terms mentioned there e.g. two-dimensional phase boundary; what is the exact role of the components of the system? what is the active site of the catalyst and so on.

Author reply:

Thanks for the reviewer pointing out the questions. The catalytic mechanism of this system can be described as follows. The hydrogen molecules dissociated on the surfaces of the TiH1.971 nanoparticles and then the hydrogen atoms could be easily transferred to the surface of Mg particles to generate MgH2 during the hydrogenation process, in this way, the activation energy for hydrogenation was reduced and the hydrogenation kinetics was enhanced. Similarly, TiH1.971 nanoparticles also effortlessly took up hydrogen atoms from the MgH2 matrix to form hydrogen molecules in the dehydrogenation reaction. As a result, the dehydrogenation kinetics of MgH2 was greatly improved after being catalyzed by TiH1.971 nanoparticles. We have added this part into the revised manuscript.

3) Page 1: Td = 300 °C, but then we read Td = 20-100 °C. I guess this should be Ta (from absorption).

Author reply:

Thanks for the reviewer’s comment. Td = 300 °C in page 1 is isothermal hydrogen dehydrogenation temperature in this paper. Td = 20-100 °C is the ideal hydrogen evolution temperature after adding catalyst as described in the reference. They are the dehydrogenation temperatures under two different systems, and there is no connection between them. In order not to confuse each other, we have deleted the description of this literature in the revised manuscript.

4) In some places '°C ' (for temperature) are used and in other data we see 'K'. Can the authors convert the values and enter the same unit everywhere?

Author reply:

Thanks for the reviewer’s careful examination. We have converted the values and provided the same unit in the revised manuscript.

5) The experimental section:

The second paragraph (MgH2 synthesis): experimental equipment should be described in more detail (scheme of the experimental setup?). Are the authors sure that the obtained samples were always the same? The fifth paragraph: 'the dates were collected' - unclear; 40 Kelvin Volt?
There is no description to the experimental setup for de-/hydrogenation reaction. Most information we read in the result and discussion section (a little bit too late).

Author reply:

Thanks for the reviewer pointing out this question.

The MgH2 powders were synthesized by heat treatment and mechanical ball Firstly, the Mg powers were heatd by a sieverts-type volumetric apparatus at 380 ºC and a hydrogen pressure of 65~70 bar. Then, the samples were ball-milled at a speed of 450 rpm for 5 h in a planetary ball mill system (QM-3SP4, Nanjing). The ball-to-powder ratio (by weight) was 40:1. Subsequently, repeating the above two steps. Finally, the products of MgH2 were obtained after absorbing hydrogen under 380 ºC and a hydrogen pressure of 65~70 bar. We cannot guarantee the MgH2 prepared each time was identical. We prepared multiple batches of MgH2 and mixed them evenly to deal this problem. 40 Kelvin Volt referred to the experimental conditions for XRD testing. The data we collect was the value of diffraction peak generated by the device. The de-/hydrogenation reaction in this paper was occurred in a high pressure of gas absorption and desorption tester designed and assembled independently by laboratory. The data of the sample’s temperature and pressure changes would be recorded by the computer. We have adjusted and transformed the relevant expressions in the revised manuscript.

6) Results and discussion section

The first paragraph (point 3.1.): XRD method may not 'see' impurities due to a small amount or non-crystalline form (so this is not the proof demonstrating the purity of the catalyst); what about using XRD method instead of TEM to determine mean size of nanoparticles in the samples? TEM may not may not distinguish agglomerates from single nanoparticles. The second paragraph (point 3.2.): the reader may be surprised finding out that the effect of catalyst concentration was also studied (there was nothing about it in the abstract and experimental section); the influence of a diameter is hardly visible in Fig. 2a and b; can you change the X-axis scale? What was the diameter after the reactions? (I think this was the real diameter of the nanoparticles - not before the temperature processes; especially when you mentioned crystal growth process e.g. on Page 7).

Author reply:

Thanks for the reviewer pointing out this question.

The raw materials used in the preparation of the catalyst are only TiH971 powders and organic solvents, in addition, the preparation process is strictly sealed. If we want to further test the purity with HRTEM measurements, we need time to make an appointment. We are sorry that we can't finish it within 5 days. The crystal size can be calculated from XRD data. However, the TiH1.971 particles may be composed of multiple crystals, so it is inappropriate to test the mean size of nanoparticles by XRD measurement. We have added an introduction to the effects of catalyst concentration in the abstract and experimental sections, and related adjustments have been made to the X-axis scale of Fig. 2(a) and (b) in the revised manuscript. This paper focuses on the hydrogen absorption and desorption performances of MgH2 catalyzed by TiH971, and its catalytic mechanism will be carefully studied in subsequent work. Therefore, the diameter after the reaction is not mentioned here. 7) Page 5: 'the date of the experiment was reliable' - unclear.

Author reply:

Thanks for the reviewer’s comment. Fig. 3(b) and (d) normalize the hydrogen absorption curves by dividing the experimental hydrogen release from the ideal hydrogen containing (7.6 wt%). If the obtained result is close to 1, it indicates that the samples reach the saturated hydrogen absorption amount and the experimental data is effective and reliable. In order to facilitate understanding, we have changed the relevant expression in the revised manuscript.

8) Page 6, Fig. 3: why do we have different temperatures for both systems (with and without catalyst) here? These two systems should be investigated in the same ranges of all parameters (the same with Fig. 4).

Author reply:

Thanks for the reviewer’s comment. The dehydrogenation temperature of MgH2 would decrease after adding the catalyst. The pure MgH2 could not desorb hydrogen in the temperature range while the composite released hydrogen rapidly. Therefore, the MgH2 was dehydrogenated at higher temperatures for the convenience of research.

9) Page 6:

The first paragraph: this should be mentioned in the abstract, introduction (to emphasize the novelty of the paper) and conclusions. Next: (t/t5)exp to (t/t0.5)theo relation is mentioned while in figures 4 we see the opposite relation. Model equations and table 1: at what assumptions can we describe chemical reactions using kinetic models for isothermal desorption? Is the adsorption (or desorption) the rate limiting step of the studied reactions?

Author reply:

Thanks for the reviewer’s comment.

The abstract, introduction and conclusion had been modified in the revised manuscript. We are so sorry that we made a mistake of writing the relationship of (t/t5)theo versus (t/t0.5)exp in the paper. The linear relation graph was produced by drawing the (t/t0.5)theo versus the (t/t0.5)exp. The descriptions of text and image had been unified in the revised manuscript. From the related literature cited in the paper, the kinetic models have been widely used in solid state reactions. The hydrogen absorption and desorption reaction we studied is a solid state reaction. So, we think we can describe the hydrogen absorption and desorption reaction by the kinetic models. A variety of factors limit the hydrogen absorption and desorption steps, including the dissociation of hydrogen, the diffusion of hydrogen atoms, and the movement of the desired interface. For Mg powers, hydrogen molecules first dissociate into hydrogen atoms on the surface of Mg, and then hydrogen atoms gradually diffuse through the phase interface of Mg, and finally bind to form MgH2 Similarly, the hydrogen release process is the opposite. In this paper, by adding TiH1.971 as a catalyst, more active sites are provided on the surface of Mg, which promotes the dissociation of hydrogen and the diffusion of hydrogen atoms. Thereby, the hydrogen absorption and desorption performance of MgH2 is improved.

10) Page 7: 'The suitable kinetics reaction models for pure MgH2 and MgH2+5 wt%-c-TiH1.971 systems were discovered' - who discovered? The authors?

Author reply:

Thanks for the reviewer’s suggestion. In this sentence, our words were not accurate. The suitable kinetics reaction models were discovered by the predecessors. We cited them here and found that the models were also applied to our system. So, we had changed the word of “discovered” to “applied” in the revised manuscript.

11) Page 7: there is not comment about the Fig. 4b.

Author reply:

Thanks for the reviewer’s careful examination and pointing out this question. We are sorry that we miss the comment about the Fig. 4(b). Fig. 4(b) is the time dependence of the kinetic modeling equation (b) for MgH2 at different temperatures. We have added it in the revised manuscript.

12) Page 8: ZrMn2, Ni/CMK-3, TiO2@C, Ta2O5, Li2TiO3 modified MgH2 systems not mentioned in the introduction. Why?

Author reply:

Thanks for the reviewer’s kind suggestion. We have added these systems to the introduction in the revised manuscript.

13) Conclusions: 'the best optimum' - the best usually means somehow optimum. 'nearly half lower than' - when we compare ca. 50 kJ/mol and ca. 75 kJ/mol, I would say the change (decrease of 75 to 50) is ca. 30% (increase of 50 to 75 would be 50%). Am I right?

Author reply:

Thanks for the reviewer’s careful examination and pointing out this question. Here, our words are not accurate enough. We have modified them in the revised manuscript.

14) English language - I am NOT a specialist (I'm not an English teacher), however, I encountered many linguistic errors in the text. As one example, see the sentence of the introduction: 'Energy is one of the most important motive force for national economic development, however, the increasing content of pollutants such as carbon dioxide in the atmosphere WERE(?) caused by high consumption of fossil fuels forced a(?) urgent demand for cleaner and sustainable energy resources.'. Therefore, I strongly recommend a careful reading of the text by an English native speaker to make sure that everything is OK. Without this it is very difficult to clearly assess the quality of the work (ideas, results, conclusions are probably publishable though).

Author reply:

Thanks for the reviewer’s careful examination and kind suggestion. The sentence mentioned by the reviewer has been carefully revised and the language of the entire article has been carefully checked again to make sure such errors do not show up.

We appreciate for reviewer’s warm work earnestly, and hope that the correction will meet with approval.

Once again, thank you very much for your comments and suggestions.

Round 2

Reviewer 1 Report

The paper has been notably improved. However, there are minor mistakes that must be corrected:

1 - The Mg-H bond is very stable and difficult to break due to the thermal stability...

Please, replace the word "thermal" for "thermodynamic".

2 - Page 8, section 3.3:

...MgH2 with and without TiH1.971 were showed in Fig....

The grammatical structure is not as correct. It would be ...are shown...

it need  English editing.

Author Response

The amendment and rebuttal list for the manuscript of

Nanomaterials-584373

The main corrections in the paper and the responds to the reviewer’s comments are as flowing:

Reviewer #1:

The paper has been notably improved. However, there are minor mistakes that must be corrected:

1 - The Mg-H bond is very stable and difficult to break due to the thermal stability...

Please, replace the word "thermal" for "thermodynamic".

Author reply:

Thanks for the kind comment. The word "thermal" has been replaced as "thermodynamic" in the revised manuscript.

2 - Page 8, section 3.3:

...MgH2 with and without TiH1.971 were showed in Fig....

The grammatical structure is not as correct. It would be ...are shown...

it need  English editing.

Author reply:

Thanks for the kind suggestion. The sentence has been rewritten in the revised manuscript. In addition, we also tried our best to check the whole paper to improve its quality.

We appreciate for reviewer’s warm work earnestly, and hope that the correction will meet with approval.

Once again, thank you very much for your comments and suggestions.

Reviewer 2 Report

Even though the authors tried to answer the questions raised in the first review report, in my opinion the present form of the paper still requires much effort to be undertaken before the paper in question may be published.

(the numbers correspond to the questions from the previous review)

1. Even if the paper deals with a hydrogen storage issue instead of sorption or hydrides formation and decomposition, there must be a strict distinction of these concepts in the text (even though the authors are sorry). The question of crystallinity is not solved as well. Choosing 'nanoparticle' rather than 'nanocrystals' is not enough. What is your definition of nanoparticles? Are these single crystals or agglomerates, or aggregates of smaller particles? How do you know you deal with nanoparticles and not with nanocrystals? Can nanoparticles be nanocrystals or not? If you had not had nanocrystalline form of particles you would have not got XRD pattern like in Fig. 1d. Based on this result I am quite sure we deal with nanocrystals.
2. Based on the description of the proposed mechanism of the process in question one can conclude that we observe the phenomenon called in catalysis 'spill-over'. Is that true?
5. Parameters characterizing the work of the X-ray tube are current (mA) and voltage (kV). There is no such unit as KV = Kelvin Volt but the authors did not notice their error even after the hint of the reviewer.
6. I suggested to use XRD method instead of TEM to determine the mean size of nanocrystals, and not to test the purity of samples. I suppose you do not need to conduct any additional XRD measurements, because probably you have already had the results (e.g. Figs. 1d or 8). You answered 'However, the TiH1.971 particles may be composed of multiple crystals, so it is inappropriate to test the mean size of nanoparticles by XRD measurement'. As I stated before, TEM may not distinguish agglomerates from single nanoparticles (nanocrystals) which can be of porous structure. Therefore, to characterize the surface (you deal with surface phenomena) and structure properties of the samples we need XRD to determine mean size of nanoparticles and BET to measure specific surface area. The size of the particles as such (visible under a microscope) is not as important as their nanostructure (studied by XRD and BET). Apart from that you admitted that there are (nano)crystals in the sample. Another thing is that you say that the diameter after the reaction is
not mentioned in the present paper, because this is going to be the topic of the next work. I do suggest to check the diameter after the process and present the results in the current work.
9. You say that 'A variety of factors limit the hydrogen absorption and desorption steps, including the dissociation of hydrogen, the diffusion of hydrogen atoms, and the movement of the desired interface.' This is not exactly true. Namely, I think that generally 'A variety of factors MAY limit the hydrogen absorption and desorption steps, including the dissociation of
hydrogen, the diffusion of hydrogen atoms, and the movement of the desired interface.' However, there is always only one limiting step in a given, particular case. Therefore, the question still remains as to what is the limiting step in the system under consideration. Apart from that we need to be aware of the assumptions of individual models to correctly: choose a model, interpret the results, propose a reaction mechanism and finally draw conclusions.
14. I am not convinced that language issues have been carefully and correctly resolved. There are still errors in the text, for instance:
- Page 1: 'The increasing content of pollutants such as nitrogen dioxide in the atmosphere resulted from high consumption of fossil fuels caused A(N) urgent demand for cleaner and sustainable energy resources.'
- Page 2: 'ball milling Mg and Ti pow(D)ers'; 'Firstly, the Mg pow(D)ers were'; 'heated by a (S)ieverts-type volumetric apparatus'
- Page 3: 'the samples were sealed in a custom-designed container and the dat(A) were collected'
- Page 5: 'and the experiment data WERE effective and reliable.'
- Page 10: 'which was in agreement with previous report(S) that'

Author Response

The amendment and rebuttal list for the manuscript of

Nanomaterials-584373

The main corrections in the paper and the responds to the reviewer’s comments are as flowing:

Reviewer #2:

Even though the authors tried to answer the questions raised in the first review report, in my opinion the present form of the paper still requires much effort to be undertaken before the paper in question may be published.

(the numbers correspond to the questions from the previous review)

1.Even if the paper deals with a hydrogen storage issue instead of sorption or hydrides formation and decomposition, there must be a strict distinction of these concepts in the text (even though the authors are sorry). The question of crystallinity is not solved as well. Choosing 'nanoparticle' rather than 'nanocrystals' is not enough. What is your definition of nanoparticles? Are these single crystals or agglomerates, or aggregates of smaller particles? How do you know you deal with nanoparticles and not with nanocrystals? Can nanoparticles be nanocrystals or not? If you had not had nanocrystalline form of particles you would have not got XRD pattern like in Fig. 1d. Based on this result I am quite sure we deal with nanocrystals.

Author reply:

Thanks for the reviewer’s comment. In the research area of hydrogen storage, the concept of sorption is usually used to describe the hydrogen storage in carbon materials (nanotubes) or other mesoporous materials (MOFs), it is a weak force. For hydrides formation and decomposition, it’s belong to chemical bond, so it’s a strong force. As for the concept of 'nanoparticle' and 'nanocrystals', in our point of view, nanoparticles must be composed of nanocrystals (excluding amorphous nanoparticles) and no matter the particles are single crystals or agglomerates, or aggregates of smaller particles, as long as the size is within range of nanometer, we call them nanoparticles. For our study, it is a solid state reaction, so we need more contact surface to improve the catalytic effect. The reviewer’s point that crystals in our particles are nanocrystals is right, we have calculated the crystal size according to the Scherrer equation and added this part in the revised manuscript.

Based on the description of the proposed mechanism of the process in question one can conclude that we observe the phenomenon called in catalysis 'spill-over'. Is that true?

Author reply:

Thanks for the reviewer’s comment. The concept of 'spill-over' was not created by us. For revealing the catalytic mechanism, researches worldwide had tried many different metals and their corresponding hydrides to modify the hydrogen storage properties of MgH2 and some conclusions have been made. Here follows the original text from one paper []:

Among a range of various theories on the effects of catalytic additives, there are two types of hypotheses: “hydrogen spillover” [25], [26] and “hydrogen gateway” [27]. “Hydrogen spillover” refers to a catalysis effect by which hydrogen molecules dissociate on the surface of additive particles and “spillover” to the surface of metallic particles to form a hydride. “Hydrogen gateway” is a sequence of phase transformation by which hydrogen is absorbed by the additive particles first, forms the additive's hydride phase, and then reacts with the light metal particle to form a metal hydride. Both mechanisms can be considered as catalytic processes. The key difference between the two theories is that the “hydrogen gateway” mechanism involves the formation of an intermediate phase, while the spillover theory does not.

In our case, the TiH1.971 stayed steady during the dehydrogenation and rehydrogenation process, indicating a mechanism of “hydrogen spillover”.

Parameters characterizing the work of the X-ray tube are current (mA) and voltage (kV). There is no such unit as KV = Kelvin Volt but the authors did not notice their error even after the hint of the reviewer.

Author reply:

Thanks for the reviewer’s careful examination. We have got this point.

I suggested to use XRD method instead of TEM to determine the mean size of nanocrystals, and not to test the purity of samples. I suppose you do not need to conduct any additional XRD measurements, because probably you have already had the results (e.g. Figs. 1d or 8). You answered 'However, the TiH1.971 particles may be composed of multiple crystals, so it is inappropriate to test the mean size of nanoparticles by XRD measurement'. As I stated before, TEM may not distinguish agglomerates from single nanoparticles (nanocrystals) which can be of porous structure. Therefore, to characterize the surface (you deal with surface phenomena) and structure properties of the samples we need XRD to determine mean size of nanoparticles and BET to measure specific surface area. The size of the particles as such (visible under a microscope) is not as important as their nanostructure (studied by XRD and BET). Apart from that you admitted that there are (nano)crystals in the sample. Another thing is that you say that the diameter after the reaction is not mentioned in the present paper, because this is going to be the topic of the next work. I do suggest to check the diameter after the process and present the results in the current work.

Author reply:

Thanks for the reviewer’s comment. As question 1, we admit that nanocrystals exist in our particles and we have modified this part in the revised manuscript. For the question of diameter of particles after the reaction, we did carry out TEM measurement last time but we got poor images as follows. 

You say that 'A variety of factors limit the hydrogen absorption and desorption steps, including the dissociation of hydrogen, the diffusion of hydrogen atoms, and the movement of the desired interface.' This is not exactly true. Namely, I think that generally 'A variety of factors MAY limit the hydrogen absorption and desorption steps, including the dissociation of hydrogen, the diffusion of hydrogen atoms, and the movement of the desired interface.' However, there is always only one limiting step in a given, particular case. Therefore, the question still remains as to what is the limiting step in the system under consideration. Apart from that we need to be aware of the assumptions of individual models to correctly: choose a model, interpret the results, propose a reaction mechanism and finally draw conclusions.

Author reply:

Thanks for the reviewer’s comment. The limiting step part has been revised as follows:

The suitable kinetics reaction models for pure MgH2 and MgH2+5 wt %-c-TiH1.971 systems were applied to the isothermal dehydrogenation tests. Fig. 4(a) showed the relationship of (t/t0.5)theo versus (t/t0.5)exp for pure MgH2 at 375 °C and the fitted linear slopes of the nine dynamics models were also listed in the picture. The A2 model had a best linear relationship due to its slope was 0.9992, which was very close to 1. Thus the nucleation and growth model of A2 (Avarami-Erofe'ev) fited well with the kinetic data of synthesized molecule of MgH2. The kinetic model changed from A2 to R2 (see in Fig. 4(c)) after adding TiH1.971 nanoparticles, indicated that the isothermal dehydrogenation process of MgH2+5 wt %-c-TiH1.971 composite was controlled by the two-dimensional phase boundary model. Moreover, isothermal dehydrogenation curves of pure MgH2 and MgH2+5 wt%-c-TiH1.971 composites at other temperatures were all well interpreted by A2 and R2 models (Fig. 4(b, d)), demonstrating these kinetics models could truly explain the dehydrogenation process. As Ti has the medium electronegativity between Mg and H (Ti (1.5), Mg (1) and H2 (2)), Ti ions are easier to gain or lose electrons (e- ) than Mg ions or H- ions. In addition, the ball milling process created a favorable contact between the TiH1.97 and MgH2. Hence, TiH1.971 could act as an intermediate carrier during the electron transferring between Mg2+ and H-. Besides, the particle size of the composite after ball milling was in the range of nanometers,[17] which would of course reduce the hydrogen diffusion distance. It was also proved that the nucleation and crystal growth process were not controlled by intraparticle diffusion but via the surface conversion of MgH2.[45] In our case, the abundance of polymorphic states of MgH2 and their slow interphase boundary migration might affect the dehydrogenation kinetics,[41] making Mg-MgH2 phase boundary movement the rate limiting step of the isothermal decomposition process in the MgH2+5 wt %-c-TiH1.971 composite under current experimental conditions.

I am not convinced that language issues have been carefully and correctly resolved. There are still errors in the text, for instance:

- Page 1: 'The increasing content of pollutants such as nitrogen dioxide in the atmosphere resulted from high consumption of fossil fuels caused A(N) urgent demand for cleaner and sustainable energy resources.'

- Page 2: 'ball milling Mg and Ti pow(D)ers'; 'Firstly, the Mg pow(D)ers were'; 'heated by a (S)ieverts-type volumetric apparatus'

- Page 3: 'the samples were sealed in a custom-designed container and the dat(A) were collected'

- Page 5: 'and the experiment data WERE effective and reliable.'

- Page 10: 'which was in agreement with previous report(S) that'

Author reply:

Thanks for the reviewer’s careful examination. We have tried our best to correct the language issues and hope that the correction will meet with approval.

Once again, thank you very much for your comments and suggestions.

Round 3

Reviewer 2 Report

Indeed, the authors have done a lot of work and the publication now looks much better than it did at first. It is a pity that much of the useful information was included by the authors in the answers to my questions No. 1 and 2. Asking these questions I did not expect any answers for myself only, because I knew these answers due to my professional expertise. I thought rather about readers that they may need this information. If the authors include in the article more information provided to me in points 1 and 2, I will recommend the work for publication.

Author Response

Author reply:

Thanks for the reviewer’s comment. All the useful information has been added in the revised manuscript. We really appreciate for reviewer’s warm work, and hope that the correction will meet with approval.

Once again, thank you very much for your suggestions.